# Relationships between fox populations and rabies virus spread in northern Canada

Susan A. Nadin-Davis[1]*, Emilie Falardeau[1¤], Alex Flynn[2], Hugh Whitney[2], H. Dawn Marshall[2]

1 National Reference Centre for Rabies, Canadian Food Inspection Agency, Ottawa Laboratory Fallowfield, Ottawa, Ontario, Canada, 2 Biology Department, Memorial University of Newfoundland, St. John's, Newfoundland & Labrador, Canada

¤ Current address: Centre for Biosecurity, Public Health Agency of Canada, Ottawa, Ontario, Canada

* nadindavis@gmail.com

**Data Availability Statement:** Mitochondrial sequences of all new haplotypes identified in this study have been deposited in GenBank (Accession Numbers MT261367-MT261383). The rabies virus complete genome sequences generated in this

## Abstract

Rabies spreads in both Arctic (*Vulpes lagopus*) and red foxes (*Vulpes vulpes*) throughout the Canadian Arctic but limited wildlife disease surveillance, due to the extensive land-mass of the Canadian north and its small widely scattered human population, undermines our knowledge of disease transmission patterns. This study has explored genetic population structure in both the rabies virus and its fox hosts to better understand factors that impact rabies spread. Phylogenetic analysis of 278 samples of the Arctic lineage of rabies virus recovered over 40 years identified four sub-lineages, A1 to A4. The A1 lineage has been restricted to southern regions of the Canadian province of Ontario. The A2 lineage, which predominates in Siberia, has also spread to northern Alaska while the A4 lineage was recovered from southern Alaska only. The A3 sub-lineage, which was also found in northern Alaska, has been responsible for virtually all cases across northern Canada and Greenland, where it further differentiated into 18 groups which have systematically evolved from a common predecessor since 1975. In areas of Arctic and red fox sympatry, viral groups appear to circulate in both hosts, but both mitochondrial DNA control region sequences and 9-locus microsatellite genotypes revealed contrasting phylogeographic patterns for the two fox species. Among 157 Arctic foxes, 33 mitochondrial control region haplotypes were identified but little genetic structure differentiating localities was detected. Among 162 red foxes, 18 control region haplotypes delineated three groups which discriminated among the Churchill region of Manitoba, northern Quebec and Labrador populations, and the coastal Labrador locality of Cartwright. Microsatellite analyses demonstrated some genetic heterogeneity among sampling localities of Arctic foxes but no obvious pattern, while two or three clusters of red foxes suggested some admixture between the Churchill and Quebec-Labrador regions but uniqueness of the Cartwright group. The limited population structure of Arctic foxes is consistent with the rapid spread of rabies virus subtypes throughout the north, while red fox population substructure suggests that disease spread in this host moves most readily down certain independent corridors such as the northeastern coast of Canada and the central interior. Interestingly the evidence suggests that these red fox populations have limited capacity to maintain the

study have been deposited in GenBank (Accession Numbers MN233898-MN234056).

**Funding:** HM received a Natural Sciences and Engineering Research Council of Canada (NSERC) Discovery grant #RGPIN2014-05265 which supported this study (https://www.nserc-crsng.gc.ca) and AF was supported by an NSERC-USRA (Undergraduate Student Research Award), # 483239 - 2015. SND received funds from the Animal Health Division of the Government of Newfoundland and Labrador. The funders had no role in study design, data collection and analysis, decision to publish, or preparation of the manuscript.

**Competing interests:** The authors have declared that no competing interests exist.

virus over the long term, but they may contribute to viral persistence in areas of red and Arctic fox sympatry.

## Introduction

Rabies is a serious zoonosis considered virtually 100% lethal once clinical signs are apparent. Rabies lyssavirus (RABV), the type species of the *Lyssavirus* genus, is the most common etiological agent of the disease. Despite the small genome size (12 Kb) of this negative sense non-segmented RNA virus and its limited coding capacity of five genes [1], this neurotropic agent propagates in the central nervous system of its victim and causes significant behavioural changes, encephalopathy and eventual death [2]. In the developed world, including Canada, wildlife species remain the principal reservoir hosts, but they can transmit the virus to most other mammalian species including humans.

Rabies has been documented in the Arctic for many decades [3] with the Arctic fox (*Vulpes lagopus*) identified as the principal reservoir host although red foxes (*Vulpes vulpes*), known for many years as a rabies host in Europe [4], are also believed to play some role in disease maintenance. Previous phylogenetic studies of Arctic RABV have suggested that this lineage most likely evolved when strains circulating in central Asia spread northwards and infected Arctic foxes which then disseminated the disease in a circumpolar manner [5–7]. An outbreak of rabies recognised in northern Canada in the 1940s [8] appears to represent the incursion of the current viral types into North America. Red foxes subsequently spread the disease into many parts of southern Canada throughout the 1950s and 1960s [9, 10] resulting in the long-term establishment of fox rabies in southern and eastern Ontario [11]. While rabies control efforts have now successfully eliminated the disease from most parts of Ontario [12], the persistence of rabies in fox populations in northern Canada continues to pose a threat of disease re-introduction. Indeed, over the last few decades there have been several instances of rabies spread from northern regions into the provinces of Quebec, Newfoundland and Labrador and central Ontario resulting in sporadic outbreaks of disease [9, 10, 13, 14]. Control measures currently remain impractical for northern regions due to the large geographical extent of fox habitat, the highly isolated and dispersed nature of most human communities and the resulting limitations in surveillance activities that would provide insight into rabies case incidence amongst fox populations. However the disease remains a public health concern in northern communities due to the threat of human exposure from infected wildlife and both domestic and feral/stray dogs which are often in direct contact with wildlife [15, 16].

The Arctic fox is distributed throughout circumpolar regions including most coastal areas of Alaska and Greenland, and most of northern Canada up to the northern tip of Ellesmere Island as well as the mainland as far south as 50˚ latitude and including the island of Newfoundland which can be accessed via ice floes during late winter and early spring [17–19]. This species is exquisitely adapted to the harsh northern climate; the excellent insulation qualities of its heavy white winter coat make Arctic fox pelts highly desirable. Although Arctic fox ranching and trapping are not as common as they were in the last century, some northern residents still make a living trapping this species. In Canada, Arctic foxes are considered abundant although their numbers can rise and fall dramatically due to significant variations in their primary food source, the lemming (genera *Lemmus* and *Dicrostonyx*) [17]. The red fox, sometimes referred to as the coloured fox in recognition of various colour morphs of this species, is larger than the Arctic fox and ranges throughout most of Canada and into many parts of the

United States [19]. Its northernmost range overlaps with that of Arctic foxes [19, 20]. Due to global warming and increased food availability in northern climates, the red fox has extended its range northwards in recent times thereby placing it in increasingly direct competition with the Arctic fox. Indeed, in Sweden and neighboring countries, decline of Arctic fox populations may be due to competition from red foxes [21]. Continued warming trends in the Arctic will exacerbate this problem and alter the range and degree of overlap of these two fox species but the implications of such changes on rabies incidence are currently unclear with predictions that the disease incidence in areas of human habitation may increase [22] or decrease [23].

To help fill gaps in knowledge of fox rabies spread in northern Canada, this study has used current methods of genetic analysis to explore rabies virus evolution over several decades together with the population structure of both red and Arctic fox hosts. Combining these two datasets with historic surveillance data supports the primary role of the Arctic fox in maintaining and rapidly spreading distinct subtypes of the rabies virus while in eastern Canada distinct red fox populations spread the virus southwards via two mostly independent corridors.

## Methods

### Rabies virus samples

Samples characterised in this study were collected from northern communities across Canada between 1977 and 2017 mostly through a passive surveillance process. Rabies suspect animals, having either human or domestic animal contact, are submitted to one of two Canadian Food Inspection Agency (CFIA) laboratories, located in Nepean, Ontario, and Lethbridge, Alberta, for diagnosis using the direct fluorescent antibody test (FAT) applied to brain smears [24]. In addition, since 2010 testing of non-contact wild animals by some provincial laboratories has been undertaken using the Direct Rapid Immunohistochemical Test (DRIT) [25] with confirmation of positive results after sample submission to, and testing by, the CFIA laboratory in Nepean. Positive cases are typed to viral variant using a panel of monoclonal antibodies [26] and brain material from such cases is maintained in long-term storage at -80˚C. Additional samples collected between 1989 and 2008 and provided by the State Virology Laboratory of the Department of Health and Social Services in Fairbanks, Alaska, were also examined. Full details of all isolates are provided in S1 Table.

### RNA extraction and viral genome amplification

RNA was extracted from brain tissue using TRIzol according to the standard procedure provided by the supplier or, for those samples to be subjected to whole genome sequencing (WGS), by a hybrid method in which the aqueous phase from the TRIzol extraction was further purified using an AMBION 1830 RNA extraction kit with a MagMax 96 deep well system as described [27]. Extracts were quantified spectrophotometrically using a Nanovue system. Complete N gene products were generated by reverse transcription-polymerase chain reaction (RT-PCR) of 2 µg RNA performed as described previously [28]. Amplification of whole viral genomes was accomplished by generation of overlapping RT-PCR products using primers adapted for the Arctic lineage RABV as described [29].

### Rabies virus sequencing and phylogenetic analysis

N gene products were sequenced on both strands using internal primers with a BigDye® Terminator v3.1 cycle sequencing kit (Applied Biosystems). Reactions were analyzed on a 3500xl genetic analyzer (Applied Biosystems) and Variant Reporter v1 software was used to assemble multiple reads into consensus sequences which were exported in fasta format. WGS was

achieved using Illumina technology; amplicons from each individual sample were quantified, pooled and processed in batches of 96 using a Nextera XT DNA library kit and normalised libraries were run on a MiSeq instrument using either 2x250 or 2x300 run kits. [27]. The resulting paired-end fastq files were subjected to a reference-based assembly using the DNASTAR Lasergene v14 software package (Madison, WI) and a reference sequence (NL.2012.0215RFX, NCBI Accession # KU198473) described previously [30]; consensus sequences were exported in fasta format.

MEGA version 7 software [31] was used for sequence alignments and phylogenetic analyses. Fasta file sets were aligned using the MUSCLE algorithm and the modeltest function was used to identify the best nucleotide substitution model for each dataset. The N gene sequence data were analysed by both Neighbour Joining (NJ) and Maximum Likelihood (ML) methods. NJ analysis employed the Maximum Composite Likelihood method to compute evolutionary distances and nodal support was determined using 1000 bootstrap replicates of the data. The ML phylogeny was generated using the Tamura-3 parameter substitution model with a discrete gamma distribution to model evolutionary rate differences among sites and employed 100 bootstrap replicates. The outgroup employed in these analyses was collected from a human case in India, (sample IND_NNV-RAB-H, NCBI Accession number EF437215.1) and is a member of the Arctic-like RABV lineage which clusters most closely to the Arctic lineage in global studies of rabies virus phylogeny.

A time-scaled phylogeny of the WGS data was constructed using the BEAST suite of programs v1.7.5 [32] in conjunction with BEAGLE [33] and using the GTR+G+I model of nucleotide substitution with a relaxed molecular clock. Two independent runs each of 50 million Markov chain Monte Carlo (MCMC) iterations with 50,000 burnin were performed and convergence of the results verified using Tracer v 1.6 software. Results were summarised as a maximum clade credibility (MCC) tree using TreeAnnotator, v1.7.5, and visualised using Figtree v1.4 [34].

To establish the consistency of the sequences generated by the WGS approach ten samples identified in S1 Table were subjected to duplicate independent amplification and sequencing. The resulting whole genomes were analysed by pairwise alignment using the MUSCLE algorithm in MEGA v7 to score any base differences.

## Evaluation of selective pressures operating on the dataset

The 208 WGS alignment used for the BEAST analysis was edited in MEGA v.7 to remove all non-coding sequence and thereby generate a 10,800 base alignment comprising a contiguous open reading frame (ORF) of 3600 amino acids encompassing all five coding regions. This alignment was analysed for the existence of selective pressures operating on these sequences using the HyPhy software [35] of MEGA7 with automatic reconstruction of a NJ phylogeny. This same dataset was also analysed for the existence of episodic selection operating on individual sites using a mixed effects model of evolution (MEME) [36] implemented in datamonkey [37]. The phylogenies generated by both methods, which employed the GTR substitution model, were similar to that produced using the unedited file in BEAST.

## Geographic mapping

Data points of RABV subtypes were mapped using the ARCGIS v10 software with templates downloaded from the Natural Earth website [38] and included provincial and territorial boundaries illustrated using a shapefile downloaded and used with permission from the Statistics Canada website [39]. These maps include the two-letter code identifying Canadian provinces and territories thus: Alberta, AB; British Columbia, BC; Manitoba, MB; New Brunswick,

NB; Newfoundland and Labrador, NL; Northwest Territories, NT; Nova Scotia, NS; Nunavut, NU; Ontario, ON; Prince Edward Island, PE; Quebec, QC; Saskatchewan, SK; Yukon, YK. The locations from which fox samples were recovered for population analysis were mapped using the Natural Earth website.

## Fox samples

Samples from non-rabies positive foxes were collected by wildlife officials or trappers stationed across northern Canada but with a special focus on the eastern half of the country where southwards movement of rabies has been observed on several occasions. A portion of hind leg muscle was excised and shipped on dry ice to Memorial University of Newfoundland. Samples from rabies-positive foxes, identified as described by the National Reference Centre for Rabies at CFIA, were provided to Memorial University as extracted DNA. Sample sizes are given in Table 1. For red foxes, microsatellite profiles were obtained for 162 individuals, including 24 rabies-positive animals, and mitochondrial control region sequences were obtained for 162 individuals, including 23 rabies-positive animals. For Arctic foxes, 157 animals were genotyped and sequenced, with 27 rabies-positive foxes. For both species, the sets of individuals investigated for both markers were virtually identical. Sampling localities are shown in Fig 1.

## DNA extraction

DNA was extracted from all tissue samples using the Qiagen QIAamp DNA Mino Kit (Qiagen Inc., Toronto, Canada) either at Memorial University or at the National Reference Centre for Rabies in Ottawa (rabies-positive animals). DNA samples were quantified spectrophotometrically using either a Nanodrop (Thermo Fisher Scientific Inc., Waltham, USA) or Nanovue system (GE Healthcare, Piscataway, NJ) and diluted to a concentration of 10 ng/μL prior to analysis.

## Mitochondrial DNA sequencing

A ~ 329 bp portion of the mitochondrial control region was targeted by PCR using the primers LF15926F and DLH [40]. PCR products were purified using the QIAQuick PCR Purification Kit (Qiagen) and Sanger sequencing reactions were performed on both strands using BigDye Terminator 3 chemistry (Thermo Fisher Scientific). Sequence reads were aligned and edited using Sequencher v4.9 (GeneCodes Corporation, Ann Arbor, USA) to generate a consensus control region sequence of ~321–329 bp for each fox.

## Microsatellite genotyping

Nine-locus dinucleotide microsatellite profiles were obtained for each fox using three multiplexes as follows: multiplex A, loci Co4.140, AHTh171, REN105L03; multiplex C, loci CPH3, AHT121, REN247M23; multiplex D, loci CPH9, CPH15, Co1.424. The CPH series of loci were described by Fredholm and Winterø [41] and the remainder by Molecular Ecology Resources Primer Development Consortium [42]. Multiplexes, fluorescent labels, and PCR conditions were as suggested by Goldsmith et al. [43]. For each multiplex, 25 μL reactions were prepared with 12.5μL Qiagen Type-It Master Mix, 30 ng of DNA template, and 2.5 μL primer mix. Primer mixes consisted of 200 nM of each primer with the exception that the mix for multiplex C contained 600 nM CPH3 F and R and the mix for multiplex D contained 400 nM CPH15 F and R. CPH15 did not always amplify well in the multiplex and in those cases was amplified separately. The thermal cycler profile consisted of 95˚C for 5 minutes, 35 cycles of 95˚C for 30

**Table 1. Sample sizes and collection localities of red (*Vulpes vulpes*) and Arctic foxes (*Vulpes lagopus*) used for population analysis.**

| Species | Locality | Code | N Microsatellites | N Mitochondrial DNA |
|---|---|---|---|---|
| *Vulpes vulpes* | | | | |
| | Churchill, MB | CHV | 50(2) | 50(2) |
| | Cartwright, NL | CAR | 18(0) | 18(0) |
| | Labrador City, NL | LAB | 23(12) | 22(11) |
| | North West River, NL | NWR | 25(0) | 24(0) |
| | Port Hope Simpson, NL | PHS | 18(0) | 18(0) |
| | Hebron, NL | | 1(1) | 1(1) |
| | Makkovik, NL | | 1(1) | 1(1) |
| | Nain, NL | | 2(2) | 2(2) |
| | Paulatuk, NT | | 1(1) | 1(1) |
| | Baker Lake, NU | | 1(0) | 2(0) |
| | Kuugaruk, NU | | 1(1) | 1(1) |
| | Kuujjuaq, QC | KUU | 15(1) | 16(1) |
| | Fermont, QC | | 1(1) | 1(1) |
| | Inukjuak, QC | | 1(1) | 1(1) |
| | Raglan Mine, QC | | 3(0) | 3(0) |
| | Schefferville, QC | | 1(1) | 1(1) |
| | Total *Vulpes vulpes* | | 162(24) | 162(23) |
| *Vulpes lagopus* | | | | |
| | Churchill, MB | CHL | 58(0) | 58(0) |
| | Port Hope Simpson, NL | | 1(0) | 1(0) |
| | Sachs Harbour, NT | SAC | 7(7) | 7(7) |
| | Holman, NT | | 3(3) | 1(1) |
| | Paulatuk, NT | | 1(1) | 1(1) |
| | Baker Lake, NU | BAK | 17(0) | 17(0) |
| | Igloolik, NU | IGL | 24(2) | 24(2) |
| | Rankin Inlet, NU | RAN | 16(1) | 16(1) |
| | Arviat, NU | | 1(1) | 1(1) |
| | Cambridge Bay, NU | | 2(2) | 2(2) |
| | Grise Fiord, NU | | 1(1) | 1(1) |
| | Gjoa Haven, NU | | 1(1) | 1(1) |
| | Kuugaruk, NU | | 2(2) | 2(2) |
| | Resolute Bay, NU | | 1(1) | 1(1) |
| | Raglan Mine, QC | RAG | 16(0) | 16(0) |
| | Ivujivik, QC | | 1(1) | 1(1) |
| | Kuujjuaq, QC | | 1(0) | 1(0) |
| | Puvirnituq, QC | | 1(1) | 1(1) |
| | Salluit, QC | | 1(1) | 1(1) |
| | Umiujaq, QC | | 2(2) | 2(2) |
| | Total *Vulpes lagopus* | | 157(27) | 157(27) |

*N* refers to sample size, while the number in brackets indicates the number of individuals within the sample set that were identified as rabies positive. A code is given to those localities considered in population genetic analyses. Note that CHV refers to red foxes from Churchill while CHL refers to Arctic foxes from Churchill.

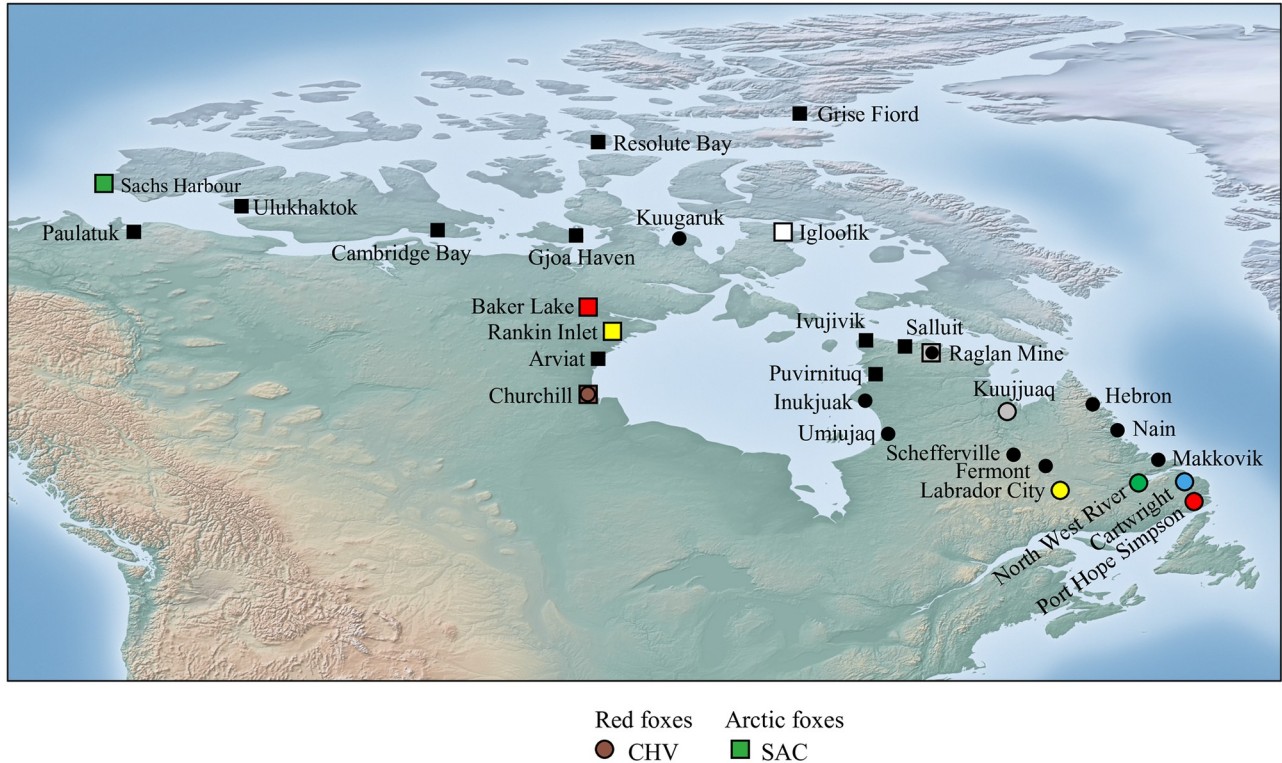

**Fig 1. Map of Canada showing localities of fox sampling for population studies.** Samples were collected between 1993–2017 with the majority recovered between 2012–2014. Each marker represents a coordinate corresponding to one or several individuals. Major sampling localities are shown by large circles for red foxes, and large squares for Arctic foxes, while those localities represented by 1–3 foxes are small circles or squares respectively.

seconds, 60°C for 45 seconds, 72°C for 30 seconds, with a final step of 60°C for 30 minutes. PCR products were diluted 1:40, and 1 μL mixed with 0.1 μL Liz Size Standard and 8.9 μL Hi Di Formamide prior to electrophoresis on an Applied Biosystems DNA Analyzer 3730 (Thermo Fisher Scientific) at the Genomics and Proteomics Facility of Memorial University. Alleles were scored using Peak Scanner v 2.0 (Thermo Fisher Scientific) and each locus for each fox was scored independently by two readers.

## Mitochondrial DNA data analysis

Arlequin v3.5 was used to determine the following measures of diversity as specified in the software: number of haplotypes, haplotype diversity and nucleotide diversity [44]; this software was also used to generate a minimum spanning tree among haplotypes for each host species. Pairwise measures of population differentiation ($F_{ST}$) were calculated in Arlequin and Spatial Analysis of Molecular Variance (SAMOVA) [45] was conducted to investigate regional structure among populations.

## Microsatellite data analysis

Potential scoring inconsistencies were assessed using Micro-Checker [46]. Linkage equilibrium was tested using Genepop 4.2 [47, 48]. Diversity measures (number of alleles, observed and expected heterozygosity) and inbreeding coefficients were calculated for loci and for populations in each species using Arlequin, as were pairwise measures of population differentiation ($F_{ST}$). Allelic richness was calculated using FSTAT [49]. SAMOVA was used to investigate spatial structure, and STRUCTURE [50] was used to assign individuals to genetic clusters. For STRUCTURE 600,000 iterations, including 100,000 burn-in, were run using the Admixture model, both with and without LOCPRIOR. Results were processed with Structure Harvester [51] and the best K was assessed using the Evanno method [52] as well as by observing the highest likelihood.

## Ethics statement

Trappers providing fox carcasses for this project are trained and licensed subject to provincial wildlife regulations. Use of the fox tissues employed for genotyping was approved by the Animal Care Committee of Memorial University under guidelines of the Canadian Council on Animal Care. Rabies positive samples were collected as part of routine diagnostic activities.

# Results

## Phylogeny of Arctic rabies viruses

To acquire as complete a picture as possible of the diversity of the Arctic RABV lineage circulating in northern regions, N gene sequences from a collection of 278 rabies positive samples (S1 Table) were compiled for alignment and phylogenetic analysis. This dataset comprised sequences of 210 Canadian samples, some of which had been sequenced previously, as well as data available in the National Center for Biotechnology Information (NCBI) repository for samples from other countries including Siberia (n = 6) [5, 53], Greenland (n = 15) [30] and Alaska, USA [5] (n = 47 including 29 sequenced during this study). A NJ tree generated from these data divided the Arctic rabies virus lineage into four sub-lineages (A1 to A4) as previously identified [5], though the A1 clade had a relatively low bootstrap value of 51%. A ML phylogeny (Fig 2) strongly supported clades A2 to A4 but divided the A1 clade, comprised of 23 samples originating exclusively from southern regions of Ontario, into two geographically segregated groups designated A1East (n = 4) and A1West (n = 19). Previous studies had identified four distinct geographically localised Ontario viral types (ON1-ON4) that have evolved independently of other members of the Arctic lineage following incursion of rabies from northern Canada in the 1950s and anecdotal information suggested that the virus which persisted in eastern Ontario (ON1), corresponding to A1East in this report, represented a separate wave of infection [54, 55]. The 15 A2 samples include five collected in the 1980s and 1990s from Siberian wildlife, eight from Arctic and red foxes of Alaska recovered between 1988–2008 and just two samples collected in the early 1990s from the northwestern region of Canada. All 27 members of the A4 clade, dated between 1989 and 2008, came from the state of Alaska. The remaining 213 samples, which were widely distributed across the north, grouped within the A3 clade.

Despite location data availability for only 20 of the Alaskan samples of this study, all viruses identified as A4 were recovered from the southern half of the state and almost exclusively from southwestern coastal regions while A2 and A3 viruses were recovered from coastal communities in the northern half of the state as reported previously [5, 23]. Indeed, it was noted that the area separating A4 viruses from those of A2 and A3 appears to run through the Bering Land

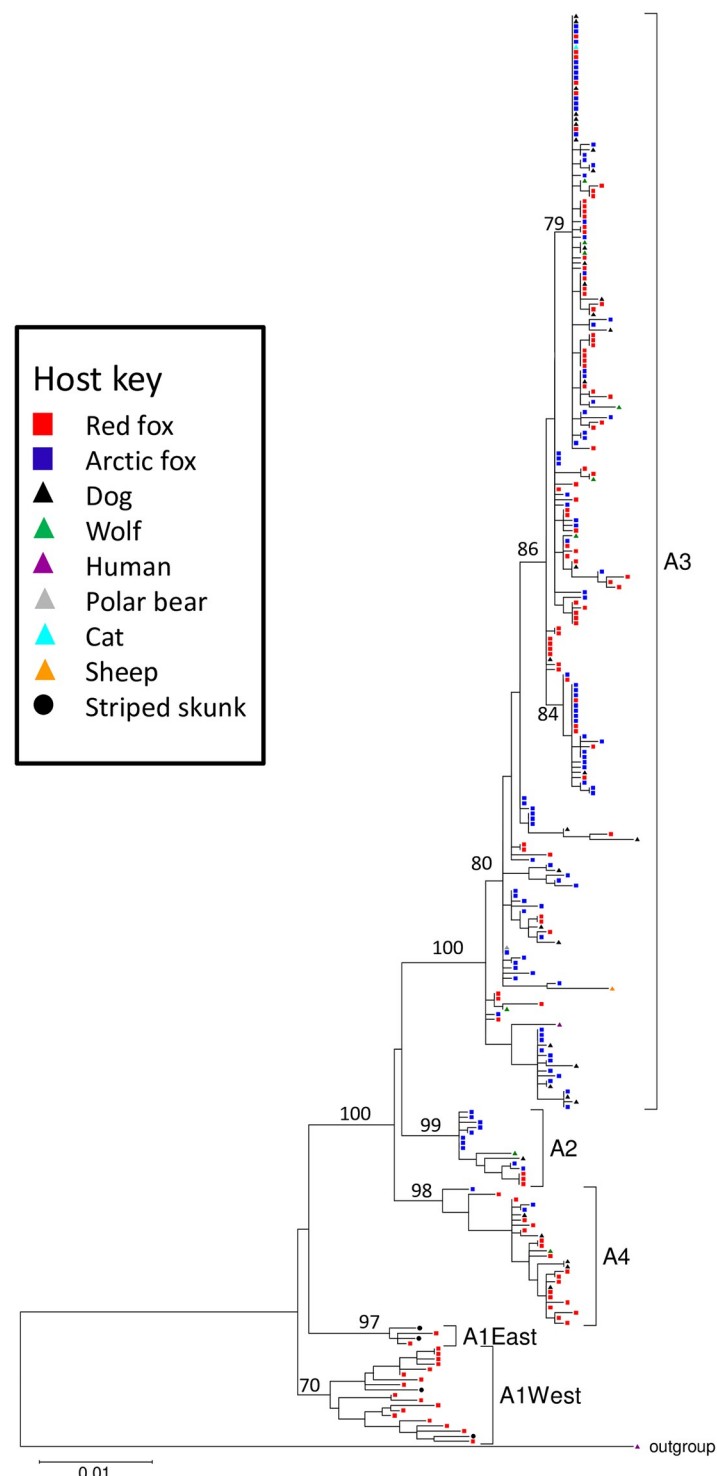

**Fig 2. Phylogeny of the Arctic rabies virus lineage using samples collected between 1977–2017.** This ML tree was generated in MEGA v7 using an alignment of 1350 positions of the N gene for 278 RABV samples and a member of the distantly related Arctic-like lineage as outgroup. Bootstrap values expressed as a percentage are indicated at major nodes. The host species from which each sample was recovered is shown according to the host key provided in the inset. The major sub-lineages are identified to the right of the tree.

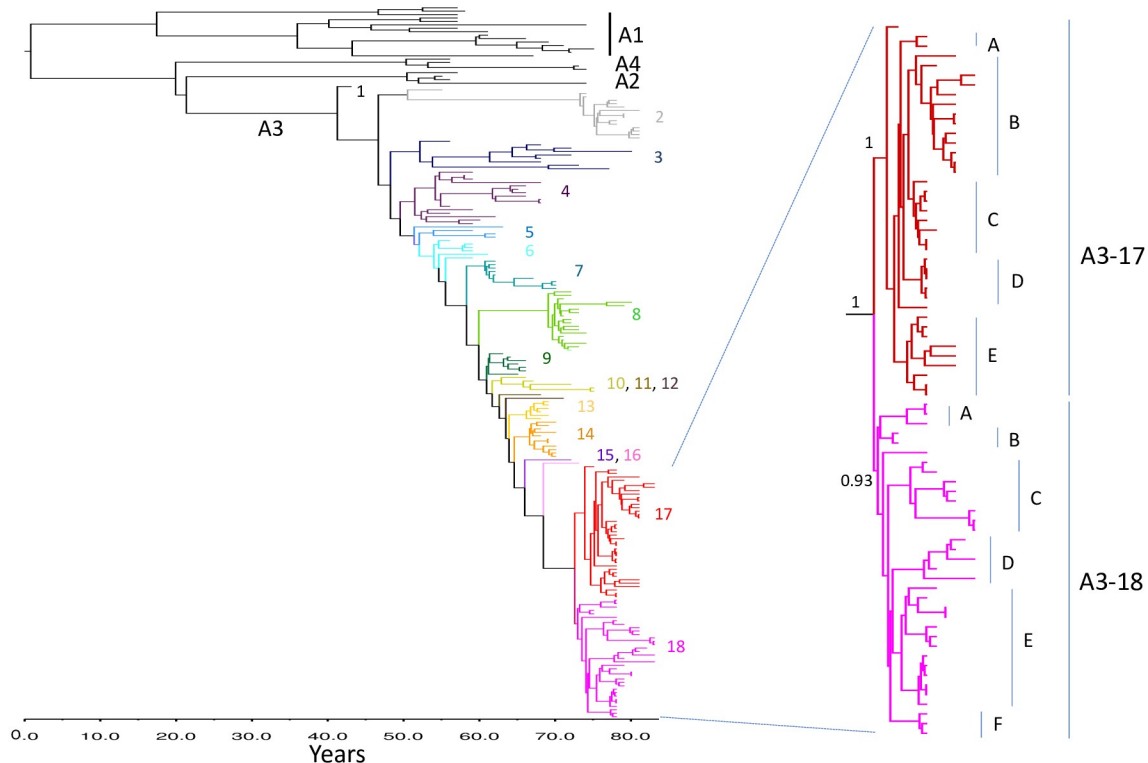

**Fig 3. A time-scaled maximum clade credibility tree produced using WGS of 208 Arctic lineage RABVs collected between 1977–2017.** The tree, which was generated using the BEAST v1.7 software package, depicts the four major sub-lineages (A1 to A4). Distinct A3 groups are colour coded and identified to the right. The predicted evolutionary time scale is shown below the tree. The inset to the right further resolves the A3-17 and A3-18 groups into several sub-groups each of which is supported by a posterior probability of ≥ 0.99.

Bridge Reserve Park, which is in closest proximity to the Asian continent. This geographic feature may have facilitated movement of rabies between these two land masses with different viral progenitors spreading into distinct regions of Alaska thereby establishing these separate disease foci over time.

While the N gene analysis did appear to suggest additional structure within the A3 clade, many of the branches did not reach a level of robustness (>70%) that allowed significant inferences to be made regarding their evolution. Accordingly, further phylogenetic analysis of the Arctic lineage, with a focus on the A3 sub-lineage responsible for all recent cases in northern Canada, was performed using WGS data for a subset of these samples. To confirm the accuracy and consistency of the sequence data generated from this WGS protocol, complete sample analysis, from amplicon generation to library construction and assembly of consensus sequences, was performed in duplicate for ten samples as identified in S1 Table. The resulting genome sequences were compared by pairwise alignment. Apart from a G/A ambiguity at base 25, introduced due to degeneracy of the amplification primer, only one other non-conserved base was identified in one sample, QC.2015.0068RFX, for which base 6663 was identified either as a T or as ambiguous T/C. These results confirm the high level of accuracy of this sequencing approach as recently reported elsewhere [56].

Fig 3 illustrates a time-scaled phylogeny generated using WGS data for the 208 samples available for this extended analysis. The much greater level of diversity inherent in these complete genome sequences clearly affords a more robust and detailed evaluation of the evolution

of these viruses. Based on this phylogeny, the Arctic lineage evolved from a single progenitor 82.5 years prior to the most recent sample included in the analysis (2017), thereby placing the year of emergence at 1935 (95% HPD range 1926–1943). Interestingly, the southern Ontario samples are all grouped with high posterior probability (>0.99) to a single A1 clade that emerged in 1951 (95% HPD range 1943–1959) with subsequent further separation into east and west groups that were similarly supported. The progenitor of the A4 group branched away in 1954 (95% HPD range 1947–1960) while the A2 and A3 sub-lineages diverged around 1955 (95% HPD range 1949–1961). All A3 sub-lineage viruses appear to have evolved from a progenitor dated to 1975 (95% HPD range 1974–1977), resulting in the evolution of multiple distinct groups, designated A3-1 to A3-18, with further diversification of the two most recently emerged groups, A3-17 and A3-18.

The distribution of 181 samples of the A3 RABV sub-lineage, which appears to be the only sub-lineage currently circulating across Canada and western Greenland, is illustrated for three time periods by the maps presented in Fig 4. Four samples of this sub-lineage, including three from Alaska, one A3-2 case and two A3-8, and a single A3-1 Canadian case are not included due to limited location data. These data suggest a complex disease transmission pattern in which only a few groups (#s A3-2, A3-4 to A3-7 and A3-9) were recovered in the 1990s with additional groups emerging in the early years of the 21st century. Notably, however, most of these viral groups have not been recovered recently in Canada and since 2010 all cases examined represent four main groups: #s A3-2, A3-8, A3-17 and A3-18, the latter two of which have become the most prominent types, apparently displacing most of the other groups identified in previous decades. Two of these groups (#s A3-3 and A3-15) appear to be restricted in range to Greenland.

The outlier of group A3-2 was recovered from a single red fox in Alaska in 1989 and subsequently it circulated, primarily in Arctic foxes, in the islands of the high Arctic throughout the 2000s. While this group continued to be recovered from the high Arctic in recent years, it had also spread south onto mainland Canada where it caused several cases, both in Arctic foxes and dogs, particularly around the eastern shore of the Hudson Bay area. Group A3-3 was first isolated from the northwestern coast of Greenland in the 1990s and probably entered the country by foxes moving over pack ice from adjacent Ellesmere Island. Since then it has spread as far as the most southern part of Greenland with cases recovered from several west coast communities. This group has not been recovered from Canada although it emerged from A3-2 viruses circulating in the northeastern Arctic region. During the 1990s group A3-4 circulated across northern Canada, mostly in Arctic foxes with cases on Baffin and Southampton islands and in northern coastal communities. This group moved south in the red fox population into Ontario and Labrador during the 2000s but has since disappeared. Group A3-5 was first identified in 1993 in an Arctic fox on the northern tip of Baffin Island, but just a few years later it was found at opposite ends of the country in both the Northwest Territories and Labrador where it was circulating in red foxes. This group and its offshoot, A3-6, which circulated widely in the north and spread into Ontario and Quebec, both disappeared before the end of the 1990s. Except for one case in Quebec, group A3-7 was recovered exclusively in the region of Labrador where it caused a significant outbreak in red foxes beginning in the 1990s and continuing into the 2000s with the last case recorded in a dog in 2004. Group A3-8, which is estimated to have emerged in 2003 was first reported in 2005 at which time it was already circulating across the northern mainland, down the western shore of Hudson Bay and on Baffin Island with cases in both red and Arctic foxes. This viral group was also recovered from Arctic foxes in both Alaska and Greenland shortly thereafter and continued to circulate with cases recovered in 2013 and 2014 in Canada and Greenland. Group A3-9 emerged in the late 1990s in Arctic foxes on Baffin Island and in an area west of Hudson Bay and by 2000 it was

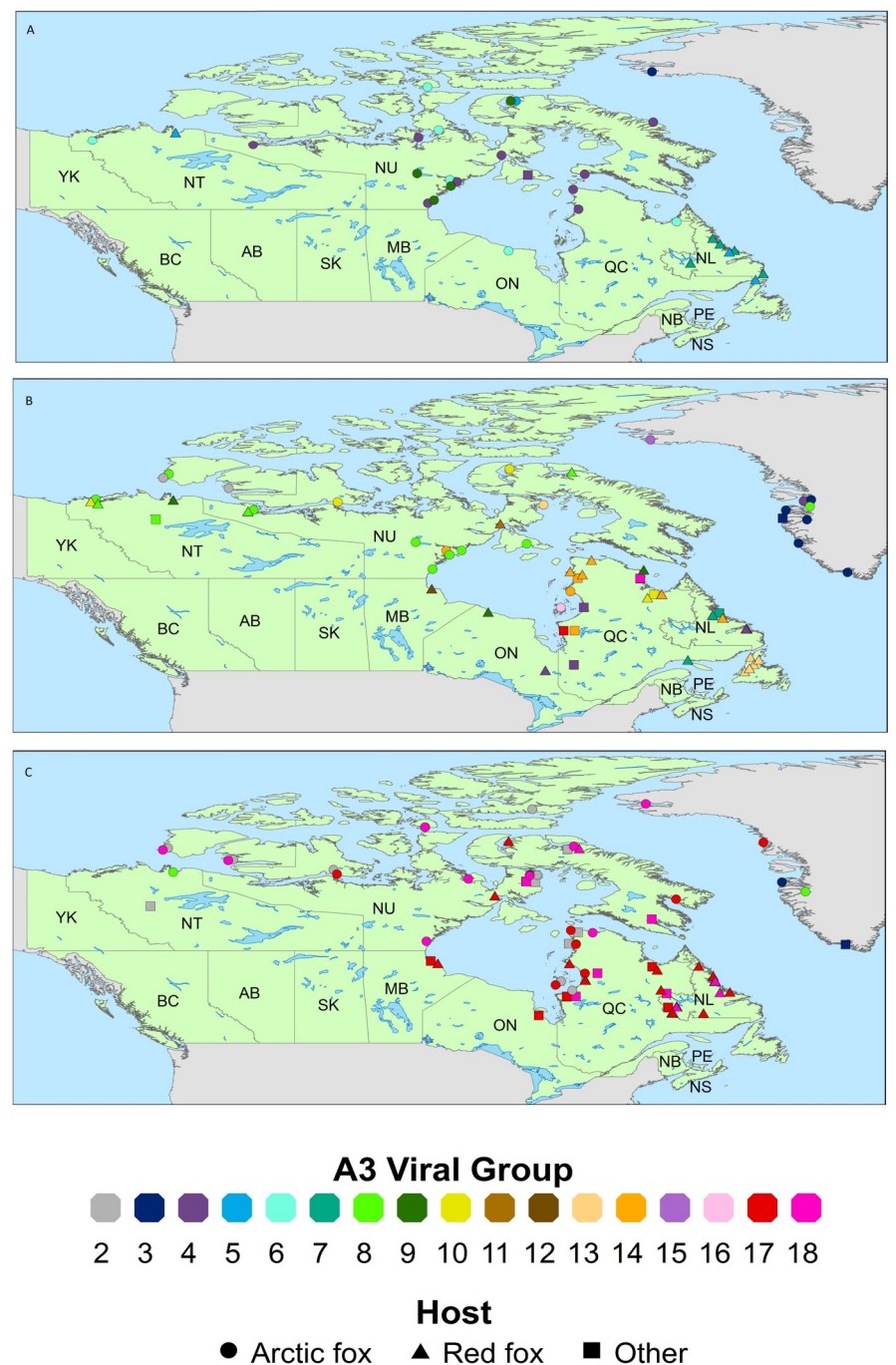

**Fig 4. Distribution of A3 sub-lineage RABVs.** Maps show the locations where viral samples of the A3 sub-lineage were recovered over three time periods: A, 1990–1999; B, 2000–2009; C 2010–2017. Points identify both the viral group, according to color, consistent with the color coding of the phylogeny shown in Fig 3, and the host species by shape as illustrated. Multiple cases of the same viral type in the same host in the same location are represented by a single point to preserve location accuracy.

also circulating in red foxes as far afield as the Northwest Territories and northern communities in Ontario and Quebec. Only five cases due to A3-10 were recovered; those from the early 2000s came from the Northwest Territories while two cases, one red fox and one wolf in 2009, were from northern Quebec. The single isolates representing A3-11 and A3-12, both from red

foxes, were from Naujaat (formerly Repulse Bay), Nunavut, and Churchill, Manitoba respectively. A single isolate of A3-13 was recovered in 2002 from an Arctic fox in Nunavut; all the other isolates of this group were recovered between 2002 and 2003 in red foxes on the island of Newfoundland. Group A3-14 was found predominantly in Quebec, especially along the eastern Hudson Bay in the 2000s; it also spread through the red fox population into Labrador by the middle of the decade before disappearing. Group A3-15 was identified by a single Arctic fox isolate from Greenland while A3-16, again identified by a single isolate, came from a small island within Hudson Bay.

The two groups which have predominated in recent years are A3-17 and A3-18 which have evolved independently from a common progenitor that existed around 2007 (95% HPD range 2006–2008) although both were first recovered from dogs in different communities in northern Quebec in 2009. In the eight subsequent years both groups were recovered widely across northern Canada. Further diversification within each of these groups is apparent (Fig 3) and the distribution of each sub-group is illustrated (S1 Fig). Some of these viral types (e.g. A3-17A, B, D and A3-18A, B) are generally restricted to Quebec and Newfoundland while others (e.g. A3-17C) were also recovered from northern Ontario and Manitoba. A3-17E was more widely distributed across Nunavut and the eastern shore of Hudson Bay. A3-18D was found only in Nunavut while A3-18C, E were distributed throughout the far north and along the eastern coast of Labrador. Simultaneous circulation of some of these viral sub-groups in certain areas has been observed. For example, in the interior Labrador City-Wabush region three sub-groups, A3-17D and A3-18A, E, were recovered in 2012. In contrast only A3-17B was isolated from this same area in 2015.

In consideration of the role of both fox species in spreading RABV, the distribution of all viral types between host species was examined more closely (Fig 2, Table 2). Except for the A1 viral clade which circulates outside of the Arctic fox range, all viral sub-lineages were recovered from both red and Arctic foxes, although the Arctic fox predominated in the A2 clade while the red fox was predominant in the A4 clade. This distribution may reflect the relative population densities of these two host species in the areas where these viral sub-lineages circulate. Within the A3 sub-lineage a few RABV groups appeared to circulate primarily in an area harbouring only one fox species (A3-2, A3-3 in Arctic foxes and A3-7 in red foxes) but most viral groups were found in areas sympatric for both fox species; this encompassed all northern coastal areas of the Canadian mainland, extending southwards to include Nunavut, most of the Northwest Territories and the northernmost areas of the provinces of Manitoba, Ontario, Quebec and Newfoundland & Labrador as well as Baffin Island. Within this vast area viral groups were usually recovered from both species except for A3-6 which was recovered only in Arctic foxes despite circulating around Hudson Bay. Indeed, there were a few examples within the sample collection in which the same RABV group was found in both species within the same community at similar times (cf. A3-8 in NT.2006.0122RFX and NT.2006.1360AFX from Tuktoyaktuk, Northwest Territories and in NU.2005.0110RFX and NU.2005.0111AFX from Kugluktuk, Nunavut; A3-17 in QC.2012.0133AFX and QC.2012.0380RFX from Umiujaq, Quebec).

Evaluation of the coding regions of the viral genome showed that mutations were due primarily to synonymous base substitutions thus supporting the wide-scale operation of purifying fixation typical of rabies viruses [57]. Using the concatenated ORF alignment of the complete WGS database, the HyPhy method did not identify any sites exhibiting evidence of positive selection at a level of P<0.05. An analysis of this same alignment using the MEME approach, which has been shown to be more sensitive in identifying potential positive episodic selection pressures, did identify nine individual sites for which positive selection was supported at the P<0.1 level. Review of the amino acid residues at these nine sites (S2 Table) showed that for the

**Table 2. Host species distribution of rabies virus types.**

| Sub-lineage | Group | Arctic fox | Red fox | Other wildlife[1] | Domestic animals[2] | Human | Total |
|---|---|---|---|---|---|---|---|
| A1 | | 0 | 19 | 4 | 0 | 0 | 23 |
| A2 | | 10 | 3 | 1 | 1 | 0 | 15 |
| A3 | 1 | 1 | 0 | 0 | 0 | 0 | 1 |
| A3 | 2 | 9 | 1 | 0 | 5 | 0 | 15 |
| A3 | 3 | 7 | 0 | 0 | 2 | 0 | 9 |
| A3 | 4 | 10 | 3 | 1 | 2 | 0 | 16 |
| A3 | 5 | 1 | 3 | 0 | 0 | 0 | 4 |
| A3 | 6 | 6 | 0 | 0 | 0 | 0 | 6 |
| A3 | 7 | 0 | 8 | 0 | 1 | 0 | 9 |
| A3 | 8 | 13 | 4 | 0 | 1 | 0 | 18 |
| A3 | 9 | 4 | 3 | 0 | 0 | 0 | 7 |
| A3 | 10 | 2 | 2 | 1 | 0 | 0 | 5 |
| A3 | 11 | 0 | 1 | 0 | 0 | 0 | 1 |
| A3 | 12 | 0 | 1 | 0 | 0 | 0 | 1 |
| A3 | 13 | 1 | 5 | 0 | 0 | 0 | 6 |
| A3 | 14 | 2 | 7 | 1 | 1 | 0 | 11 |
| A3 | 15 | 1 | 0 | 0 | 0 | 0 | 1 |
| A3 | 16 | 1 | 0 | 0 | 0 | 0 | 1 |
| A3 | 17 | 10 | 19 | 2 | 8 | 0 | 39 |
| A3 | 18 | 14 | 13 | 1 | 7 | 0 | 35 |
| A3 | ungrouped | 10 | 11 | 2 | 4 | 1 | 28 |
| A4 | | 3 | 18 | 1 | 5 | 1 | 27 |
| | | | | Grand total | | | 278 |

[1] For the A1 clades the striped skunk represented all other wildlife. For sub-lineages A2-A4 wolves comprised all other wildlife except for one polar bear (group A3-4)

[2] Dogs represented all domestic animals except for one ovine (group A3-3) and one cat (group A3-17).

most part they were well conserved with substitutions noted in just a small number of isolates. There was no correlation of such changes with viral subtype or group or the species of origin with one exception: all four of the A4 sub-lineage samples analysed by WGS exhibited a R to Q change at site 3517 corresponding to amino acid 2044 of the L protein. While this change may be fixed in this sub-lineage, this same substitution was also noted in two other unrelated isolates: one of group A3-17 and one of A1-ON4. Overall, however, there was no indication that coding differences within Arctic lineage rabies viruses are associated with either fox host.

## Fox population analysis

**Mitochondrial data.** Among 162 red foxes, 130 variable sites defined 18 control region haplotypes, of which 4 are new to this study (S3 Table, Fig 5). Of these, 11 were shared and 7 found in only one individual. The two most common haplotypes account for 98 of the 162 foxes (60.5%) and are found in each of the six primary sampling localities. Among 157 Arctic foxes, 26 variable sites defined 33 haplotypes (S4 Table, Fig 6). Of these 16 were shared and 17 are unique to individuals; the two most common haplotypes account for 64 of the 157 foxes (40.8%), and only the most common is found in the six primary sampling localities. 18 haplotypes had been described and 15 are new to this study.

Diversity measures for mitochondrial DNA are presented in Table 3. Number of haplotypes ranged from $N_H = 2$ in Cartwright (NL) to $N_H = 11$ in Churchill (MB) for red foxes, and from

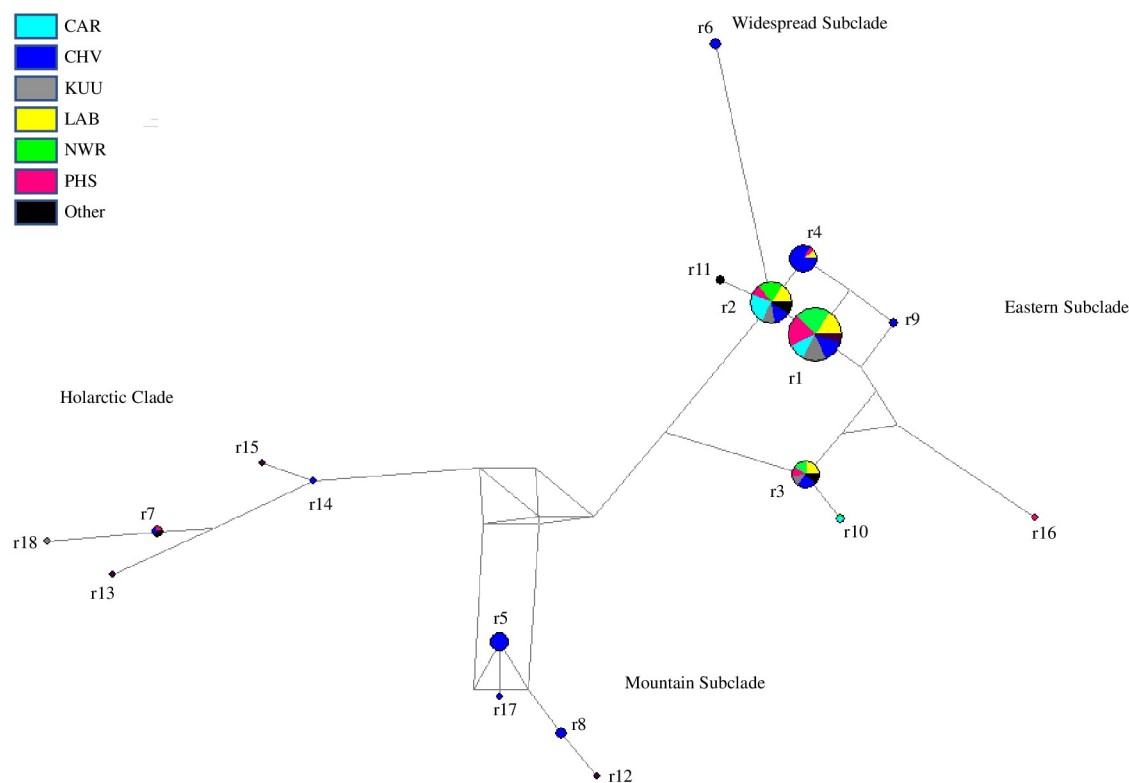

**Fig 5. Median spanning network of 18 red fox mitochondrial control region haplotypes generated using network 5.0.** Circles representing haplotypes are scaled according to frequency, with proportional membership in sampling localities as indicated in the key. Sample locality codes are given in Table 1.

$N_H$ = 5 in Rankin Inlet (NU) and Sachs Harbour (NT) to $N_H$ = 18 in Churchill (MB) for Arctic foxes; and was on average twice as high in Arctic vs red foxes. Haplotypic and nucleotide diversity were also lowest in Cartwright (NL) ($h$ = 0.529, $\pi$ = 0.0016) and highest in Churchill (MB) ($h$ = 0.861, $\pi$ = 0.0193) in red foxes, and averaged $h$ = 0.685 and $\pi$ = 0.00932 in this species. In Arctic foxes, haplotypic diversity ranged from h = 0.450 in Rankin Inlet (NU) to h = 0.949 in Baker Lake (NU), and nucleotide diversity from $\pi$ = 0.00408 in Rankin Inlet (NU) to $\pi$ = 0.0159 in Igloolik (NU), with averages of h = 0.825 and $\pi$ = 0.00998 in this species. Overall nucleotide diversity is about the same in the two species, haplotypic diversity is about 20% higher in Arctic foxes, and number of haplotypes in Arctic foxes is double that of red foxes.

Measures of population differentiation assessed with mitochondrial DNA differed between red and Arctic foxes. Average pairwise $F_{ST}$ (Table 4) in red foxes was 0.046 with the highest levels between Churchill (MB) and the other populations, and positive values between Cartwright (NL) and the other sampling locations. Values of $F_{ST}$ were negative or non-significant among Kuujjuaq (QC), Labrador City (NL), North West River (NL) and Port Hope Simpson (NL). $F_{ST}$ was generally negative or non-significant among pairs of sampling locations of Arctic foxes except for Raglan Mine (QC) and Rankin Inlet (NU) ($F_{ST}$ = 0.079, P = 0.0451, Table 5). SAMOVA (Table 6) for red foxes suggested a moderate level of regional differentiation at K = 2–5, but none was significant at P<0.05. At K = 4, $F_{CT}$ = 0.147 (P = 0.05) suggested that Cartwright (NL), Churchill (MB) and Port Hope Simpson (NL) are separate entities but that North West River (NL), Labrador City (NL) and Kuujjuaq (QC) group together. For Arctic foxes $F_{CT}$ was lower for all K values than for red foxes. The only significant $F_{CT}$ was observed

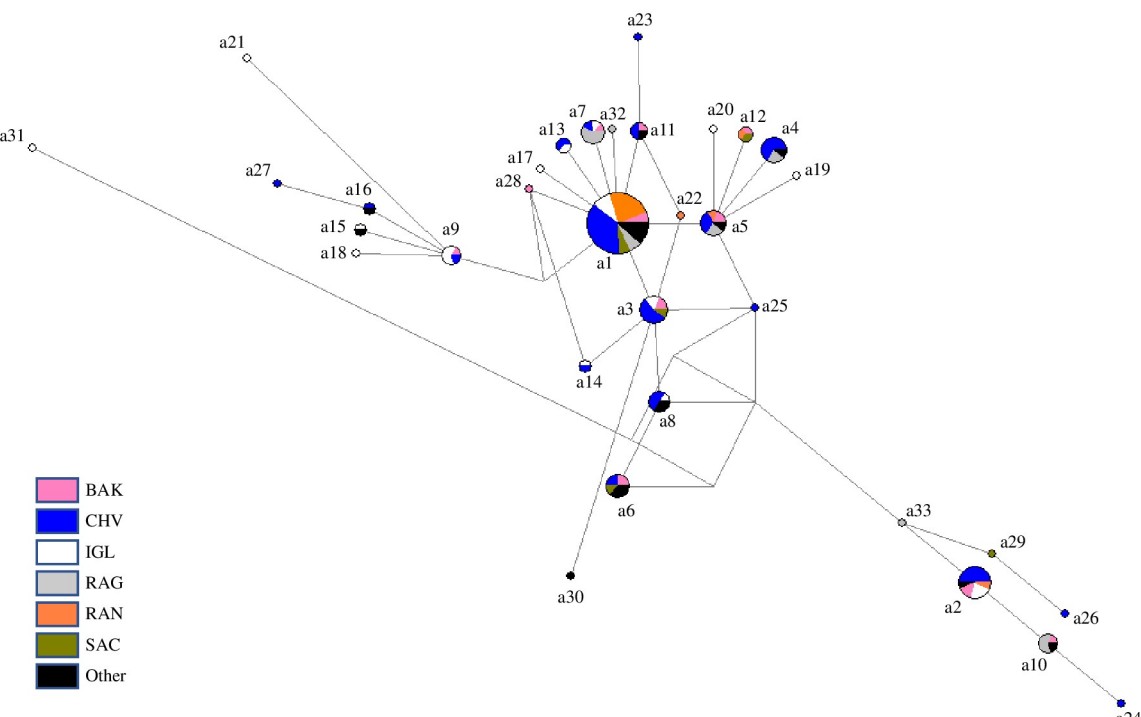

**Fig 6. Median spanning network of 33 Arctic fox mitochondrial control region haplotypes generated using network 5.0.** Circles representing haplotypes are scaled according to frequency, with proportion membership in sampling localities as indicated in the key. Sample locality codes are given in Table 1.

**Table 3. Measures of mitochondrial DNA diversity, and microsatellite diversity and $F_{IS}$, in red (*Vulpes vulpes*) and Arctic (*Vulpes lagopus*) foxes across northern Canada.**

| Locality | Mitochondrial DNA | | | Microsatellites | | | |
|---|---|---|---|---|---|---|---|
| | $N_H$ | $h$ | $\pi$ (x10⁻³) | $N_A$ | $k$ | $H_E$ | $F_{IS}$ |
| *Vulpes vulpes* | | | | | | | |
| CAR | 2 | 0.529 | 1.61 | 4.56 | 4.436 | 0.631 | -0.0215 |
| LAB | 4 | 0.710 | 6.74 | 6.00 | 5.425 | 0.665 | 0.0037 |
| NWR | 4 | 0.630 | 5.97 | 5.22 | 4.835 | 0.620 | 0.1012 |
| PHS | 7 | 0.739 | 12.1 | 5.33 | 5.112 | 0.611 | -0.0023 |
| KUU | 4 | 0.642 | 10.2 | 5.44 | 5.354 | 0.661 | -0.0679 |
| CHV | 11 | 0.861 | 19.3 | 7.44 | 5.842 | 0.695 | -0.0076 |
| Mean | 5.33 | 0.685 | 9.32 | 5.67 | 5.167 | 0.647 | 0.0056 |
| *Vulpes lagopus* | | | | | | | |
| RAG | 7 | 0.883 | 11.0 | 8.56 | 5.587 | 0.770 | 0.0825 |
| CHL | 18 | 0.871 | 9.22 | 10.0 | 5.446 | 0.757 | 0.0280 |
| BAK | 11 | 0.949 | 10.5 | 7.67 | 5.426 | 0.783 | -0.0102 |
| IGL | 15 | 0.938 | 15.9 | 8.22 | 5.295 | 0.751 | -0.0180 |
| RAN | 5 | 0.450 | 4.08 | 8.67 | 5.699 | 0.792 | 0.0099 |
| SAC | 5 | 0.857 | 9.20 | 5.78 | 5.415 | 0.744 | -0.0166 |
| Mean | 10.2 | 0.825 | 9.98 | 8.15 | 5.511 | 0.766 | 0.0126 |

$N_A$: average number of alleles; $H_E$: gene diversity; $k$: allelic richness; $N_H$: number of haplotypes; $h$: haplotypic diversity; $\pi$: nucleotide diversity. No $F_{IS}$ was significantly greater than zero at P = 0.05 after Bonferroni correction for multiple tests.

**Table 4. Pairwise measures of population differentiation ($F_{ST}$) within the red fox in northern Canada.**

| Pop | CAR | LAB | NWR | PHS | KUU | CHV |
|---|---|---|---|---|---|---|
| CAR | - | 0.075 | 0.080 | **0.113** | **0.036** | **0.155** |
| LAB | **0.049** | - | -0.037 | -0.024 | -0.034 | **0.110** |
| NWR | **0.017** | 0.010 | - | -0.019 | -0.034 | **0.134** |
| PHS | **0.074** | 0.009 | **0.020** | - | -0.045 | **0.102** |
| KUU | **0.024** | 0.003 | 0 | **0.026** | - | **0.085** |
| CHV | **0.058** | **0.025** | **0.036** | **0.035** | **0.015** | - |

$F_{ST}$ values from mitochondrial data are above the diagonal, and those from microsatellites are below the diagonal. $F_{ST}$ values significant at P = 0.05 are in bold. Locality codes are as given in Table 1.

**Table 5. Pairwise measures of population differentiation ($F_{ST}$) within the Arctic fox in northern Canada.**

| Pop | RAG | CHL | BAK | IGL | RAN | SAC |
|---|---|---|---|---|---|---|
| RAG | - | 0.008 | -0.021 | 0.035 | **0.079** | -0.026 |
| CHL | 0.010 | - | -0.025 | 0.018 | 0.027 | -0.047 |
| BAK | 0.013 | **0.012** | - | -0.010 | 0.043 | -0.080 |
| IGL | **0.015** | 0.002 | **0.014** | - | 0.034 | -0.032 |
| RAN | **0.018** | **0.016** | **0.017** | **0.025** | - | -0.020 |
| SAC | -0.005 | -0.007 | 0.011 | -0.004 | -0.003 | - |

$F_{ST}$ values from mitochondrial data are above the diagonal, and those from microsatellites are below the diagonal. $F_{ST}$ values significant at P = 0.05 are in bold. Locality codes are as given in Table 1.

for K = 4 ($F_{CT}$ = 0.052, P = 0.044), and suggested that Churchill (MB), Baker Lake (NU), and Sachs Harbour (NT) group together, with the other three localities as separate entities.

**Microsatellite data.** Micro-Checker analysis indicated a significant (P < 0.05) excess of homozygotes consistent with null alleles for two loci in two different populations (CPH3 in PHS; REN247M23 in CHV) of red foxes, and one locus (REN105L03) in one population (CHL) of Arctic foxes (S5 Table). For two of these loci (REN247M23, REN105L03) a

**Table 6. Grouping of sampling locations determined by Spatial Analysis of Molecular Variance (SAMOVA) of microsatellites or control region sequences, for red foxes and Arctic foxes.**

| K | $F_{CT}$ (P) | Grouping of Locations | $F_{CT}$ (P) | Grouping of Locations |
|---|---|---|---|---|
| | **Mitochondrial DNA** | | **Microsatellites** | |
| **Red foxes** | | | | |
| 2 | 0.178 (0.173) | CHV, others | 0.025 (0.160) | CHV, others |
| 3 | 0.164 (0.065) | CAR, CHV, others | 0.037 (0.058) | CAR, CHV, others |
| 4 | 0.147 (0.050) | CAR, CHV, NWR/LAB/KUU, PHS | 0.047 (0.052) | CAR, CHV, NWR, PHS/LAB/KUU |
| 5 | 0.135 (0.108) | CHV, CAR, NWR/LAB, KUU, PHS | 0.052 (0.066) | CAR, CHV, PHS, NWR, LAB/KUU |
| **Arctic foxes** | | | | |
| 2 | -0.054 (1.00) | RAG/CHL/IGL/RAN/BAK, SAC | 0.036 (0.071) | RAN/BAK, RAG/CHL&IGL&SAC |
| 3 | 0.043 (0.057) | RAG/CHL/BAK/SAC, RAN, IGL | 0.036 (0.013) | RAN/BAK, RAG/CHL/SAC, IGL |
| 4 | 0.052 (0.044) | CHL/BAK/SAC, RAN, IGL, RAG | 0.044 (0.018) | IGL, CHL/SAC, RAG, RAN/BAK |
| 5 | 0.081 (0.062) | BAK/SAC, IGL, CHL, RAN, RAG | 0.042 (0.065) | IGL, CHL, SAC, RAG, RAN/BAK |

K is the number of groups; $F_{CT}$ is the F-statistic indicating the proportion of variance among groups.

significant lack of heterozygotes differing by one repeat unit suggested stuttering may be occurring. As no loci or populations were consistently problematic, we proceeded to use all loci for further analysis in both species. Analysis of linkage equilibrium using Genepop 4.2 revealed two pairs of loci in linkage disequilibrium in two different populations (AHT121/REN247M23 in CAR; Co4.140/CPH3 in CHV) in red foxes (P < 0.05 after Bonferroni correction; adjusted $\alpha$ = 0.00139). For Arctic foxes, four instances of linkage disequilibrium were noted in IGL (AHTh171/CPH3, REN105L03/CPH3, CPH3/REN247M23, CPH3/CPH15), and one each in RAG (AHTh171/REN247M23) and RAN (AHT121/REN247M23). As these potential linkages do not occur within multiplexes and are not consistent across samples and species, they most likely do not reflect scoring errors or actual physical linkages. Instead, these results suggest that some level of population admixture may exist in some of the *a priori* sample designations, particularly IGL. We proceeded with analyses of all loci noting, where applicable, the influence this may have on downstream results and interpretation.

The locus statistics presented in S5 Table show that measures of diversity were higher for all loci for Arctic foxes than for red foxes. On average $H_E$ = 0.772 for Arctic foxes and $H_E$ = 0.675 for red foxes. One locus showed significantly positive $F_{IT}$ for each species, REN247M23 in Arctic and REN105L03 in red foxes. Average $F_{ST}$ was higher for red foxes with several loci exhibiting significantly positive values; no loci showed a significant $F_{ST}$ for Arctic foxes.

Population diversity measures for microsatellites (Table 3) show similar trends to those for mitochondrial DNA when comparing species but some differences within species. Both average number of alleles and expected heterozygosity were higher in Arctic ($N_A$ = 8.15, $H_E$ = 0.766) than red foxes ($N_A$ = 5.67, $H_E$ = 0.647), while allelic richness was similar for the two ($k$ = 5.167 for red foxes, $k$ = 5.511 for Arctic foxes). Cartwright (NL) showed the lowest allelic richness in red foxes, while Port Hope Simpson (NL) showed the lowest expected heterozygosity. Churchill (MB) was the most diverse sampling locality across all measures. For Arctic foxes, Rankin Inlet (NU) showed the highest diversity across all measures ($N_A$ = 8.67, $k$ = 5.699, $H_E$ = 0.792), in contrast to the pattern seen with mitochondrial DNA, where it was the least diverse sampling location. The lowest allelic richness was seen in Igloolik (NU) ($k$ = 5.295) while the lowest expected heterozygosity was in Sachs Harbour (NT) ($H_E$ = 0.744). $F_{IS}$ was not significantly positive in either species for any sampling locality.

Pairwise measures of $F_{ST}$ for red foxes (Table 4) averaged $F_{ST}$ = 0.027 with Churchill (MB) being the most highly differentiated locality (average $F_{ST}$ to other localities $F_{ST}$ = 0.044), and Cartwright (NL) the second most (average $F_{ST}$ to other localities $F_{ST}$ = 0.034). Among the remaining localities Port Hope Simpson (NL) was significantly different from North West River (NL) ($F_{ST}$ = 0.020) and Kuujjuaq (QC) ($F_{ST}$ = 0.026) but Labrador City (NL), North West River (NL) and Kuujjuaq (QC) were not significantly different from each other. SAMOVA (Table 6) suggested the greatest level of regional differentiation at K = 4 ($F_{CT}$ = 0.052) grouping Labrador City (NL) and Kuujjuaq (QC) together and each other locality separately, although this result was not significant (P = 0.066).

STRUCTURE analysis for the red fox is presented in Fig 7. Under the default no admixture model, the best supported K was K = 2 according to the Evanno approach. The likelihood continued to increase with increasing K, but not substantially after K = 3, hence results for both K = 2 and K = 3 are shown. K = 2 shows that Churchill (MB) is dominated by one genetic cluster (red), and the Labrador/Quebec locations by the other (green). Individuals with membership >50% in one cluster can be found in sampling localities dominated by the other cluster, although Cartwright (NL) contains only one such individual. K = 3 introduces a third (blue) cluster with membership in all localities, although most notably in Port Hope Simpson (NL) and North West River (NL) and less so in Kuujjuaq (QC) Labrador City (NL), and Churchill (MB). For the LOCPRIOR model the best K was selected as the K with the highest likelihood,

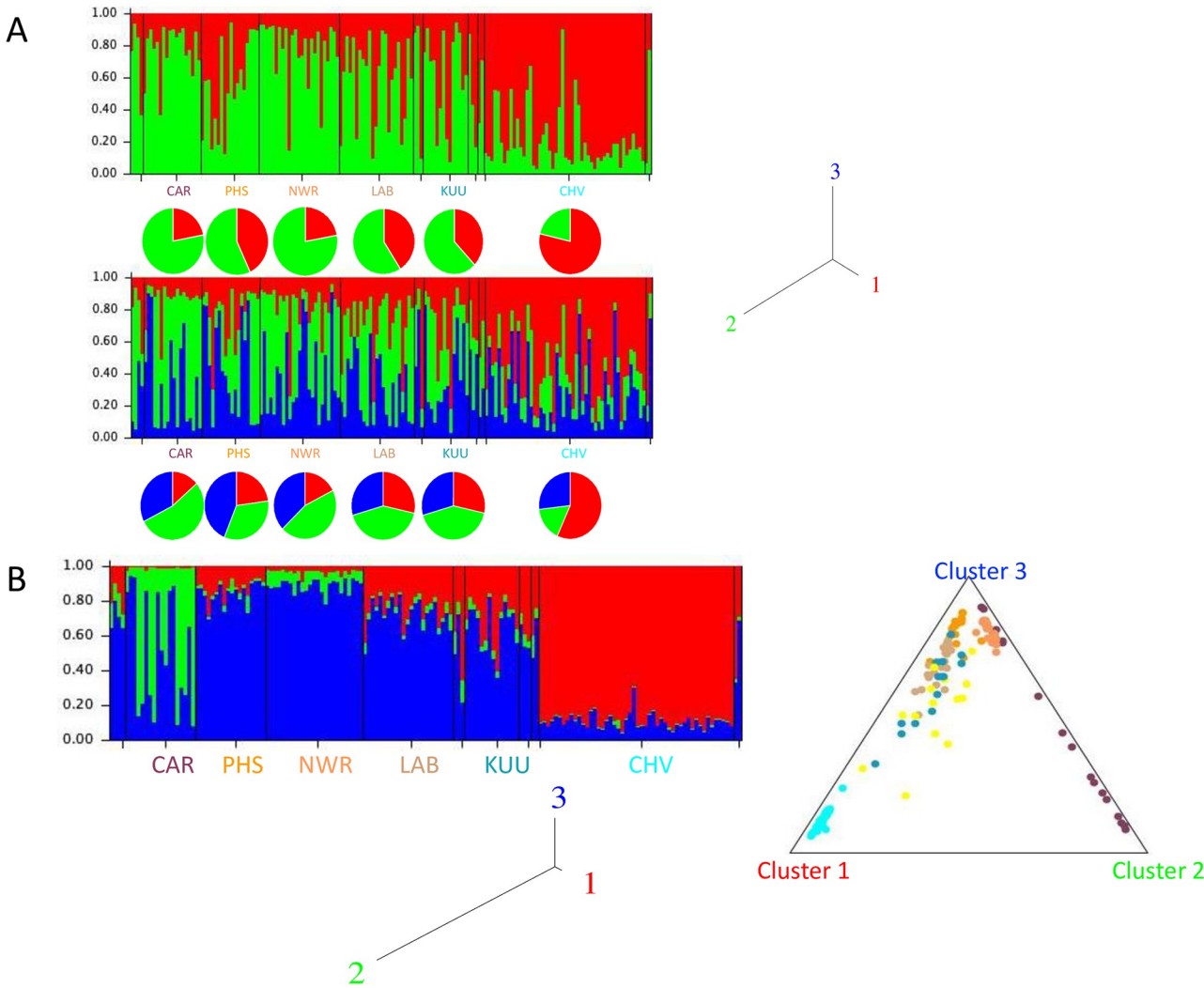

**Fig 7. Structure analysis of red foxes.** A. Plot of individual membership in K = 2 and K = 3 genetic clusters using the default admixture model with 100,000 burn-in and 500,000 retained iterations. Sample locality codes are given in Table 1. Tree plot of the relationships among the K = 3 genetic clusters under this model. B. A. Plot of individual membership in K = 3 genetic clusters using the admixture model with LOCPRIOR, with 100,000 burn-in and 500,000 retained iterations. Sample locality codes are given in Table 1. Triangle plot of individual association with each of the K = 3 genetic clusters under the admixture with LOCPRIOR model. Individuals are colour-coded as indicated in A and B. The yellow dots represent the individuals not in one of the six main sampling localities, as indicated in Table 1. Tree plot of the relationships among the K = 3 genetic clusters, under the admixture with LOCPRIOR model.

as for this analysis there was only one such K, K = 3. This result suggests Churchill (MB) contains primarily one cluster (red) as before, Cartwright (NL) consists of a mix of two clusters: numbers two (green) and three (blue). Port Hope Simpson (NL), Labrador City (NL) and Kuujjuaq (QC) individuals are admixed between the first (red) and third (blue) clusters, and the proportion of the first (red) is lowest in Port Hope Simpson (NL) and similar to the third (blue) in Labrador City (NL) and Kuujjuaq (QC). North West River (NL) contains largely individuals from the third (blue) cluster. The r value for this analysis was $r = 0.32$, suggesting that locations are informative regarding population structure.

Pairwise measures of $F_{ST}$ for Arctic foxes (Table 5) were on average $F_{ST} = 0.010$, with Igloolik (NU), Baker Lake (NU), and Rankin Inlet (NU) significantly differentiated from each other, Igloolik (NU) and Rankin Inlet (NU) also from Churchill (MB) and Raglan Mine (QC),

and Baker Lake (NU) from Churchill (MB). Sachs Harbour (NT) was not significantly differentiated from any other locality, perhaps due to its small sample size; the only positive differentiation was with Baker Lake (NU) ($F_{ST}$ = 0.011). Nonetheless, SAMOVA (Table 6) suggested significant regional differentiation at K = 3 and K = 4, with the highest level at K = 4 ($F_{CT}$ = 0.044, P = 0.018), grouping Churchill (MB) with Sachs Harbour (NT), Baker Lake (NU) with Rankin Inlet (NU), and Raglan Mine (QC) and Igloolik (NU) separately. This result differs from the mitochondrial analysis, in which Baker Lake (NU) aligned with Churchill (MB) and Sachs Harbour (NT). STRUCTURE analysis did not reveal any evidence for any distinct genetic clusters among Arctic fox individuals; with or without LOCPRIOR, K = 1 was the best supported K.

## Discussion

The persistence of Arctic rabies in the circumpolar region has been the subject of some debate for many years given that the presumed primary host reservoir, the Arctic fox, is subject to low and variable population density which will tend to restrict opportunities for disease transmission. Bias in detection and reporting of rabies cases in remote northern regions can confound our knowledge of the relative importance of a species as a rabies reservoir. This study has explored the relative importance of two fox hosts responsible for harboring rabies virus in northern areas of Canada by analysing their population structures using both mitochondrial sequencing and microsatellite analysis and this allows us to correlate these results with the evolution of the Arctic lineage of rabies virus that circulates across Canada and neighboring countries.

Our study of Arctic foxes identified a similar number and pattern of haplotype relationships to that reported by Dalén et al. [58] who found 29 haplotypes among 191 Arctic foxes distributed circumpolarly; we find 18 of their haplotypes and 15 new haplotypes. As in their study, most of our haplotypes are closely related to each other and differ by one or two nucleotide substitutions from their nearest neighbour in the network and include one new haplotype, a31, containing a 16-bp deletion relative to all other haplotypes. However, as for studies of Arctic foxes in Greenland [30], there is little evidence to support the existence of distinct Arctic fox populations across northern Canada. Indeed, the evidence supports extensive gene flow between members of this host species throughout the region no doubt facilitated by their frequent long-distance movements. This is especially notable during natal dispersal, as documented recently by satellite collar data for a young vixen who travelled an average of 46.3 kilometers a day over 76 days between the Svalbard islands and Ellesmere Island in northern Canada [59]. The ability of these animals to disperse long distances over pack ice renders them eminently capable of rapidly spreading diseases throughout the Arctic, particularly a viral disease such as rabies which can have an incubation period of weeks or even months.

The genetic diversity of the Arctic lineage of rabies virus described here clearly supports a model in which the virus is continuously evolving through mostly neutral mutation and purifying selection to spawn distinct genetic groups. Modelling studies of Arctic rabies have suggested that a variety of factors including high transmission rates, relatively short incubation periods but long infectious periods and ongoing immigration of infectious conspecifics from other areas could help in maintaining rabies outbreaks in northern Canada; however, various scenarios in which outbreak die-offs occurred were also often predicted [60]. Indeed, the rabies virus phylogeny illustrated here does suggest that while some viral groups appear to circulate only very locally and die off relatively quickly (e.g. A3-11, -12. -16) others circulate widely over several years (e.g. A3-2, -3, -8). In some situations convergence of species members on localised food sources, including anthropogenic structures such as waste dumps, and

scavenging of carcasses on ice may help to facilitate disease transmission within a relatively small population which may then disperse and spread certain viral variants widely. Such widespread Arctic fox movements would explain the relatively rapid spread of the recently emerged viral groups A3-17 and A3-18 over much of the study area. Based on our limited comparison of Canadian and Greenland samples it appears that Greenland acts mainly as a rabies sink with viral groups dispersing from Canada into Greenland but rarely in the opposite direction.

In contrast to our observations on Arctic foxes, red foxes displayed generally lower genetic diversity, but greater mtDNA nucleotide diversity, suggesting a longer evolutionary history but smaller population sizes than Arctic foxes. As well, red foxes display stronger population genetic structure, possibly reflecting both isolation into distinct forest refugia during the Pleistocene [61], as well as more limited ongoing movement than Arctic foxes, as supported in these studies. The 18 haplotypes were assigned to the Holarctic and Nearctic clades, and to the Widespread, Mountain, and Eastern subclades of the Nearctic clade, and exhibited similar distribution across Canada as previously described [61], with Labrador and Quebec predominated by Eastern subclade types and central and northwestern parts of Canada containing Mountain subclade types. One Widespread haplotype was observed in Churchill (MB), and 6 Holarctic haplotypes were present in various locations including Churchill (MB), Port Hope Simpson (NL), Paulatuk (NT), and Kuujjuaq (QC).

Both marker types suggest a similar pattern of differentiation with a Churchill (MB) vs Cartwright (NL) vs (Kuujjuaq (QC), Labrador City (NL), North West River (NL), Port Hope Simpson (NL)) pattern; Port Hope Simpson (NL) is also somewhat differentiated versus the others. Thus, there is a northwestern vs northeastern pattern of differentiation, and an interior vs coastal distinction in the northeastern region; a map of eastern Canada illustrates the approximate locations of these distinct populations (S2 Fig). The Cartwright (NL) group is comprised of foxes trapped mostly on small islands in the bay, and they are more likely to be cross or silver in colour than foxes collected from other localities. Coincidentally, fox farming activity was conducted at nearby Sandwich Bay in the 1910s involving foxes trapped at various locations along the Labrador coast [62]. The genetic distinction of the Cartwright (NL) group may result from this or from its isolated location. The STRUCTURE analysis of microsatellites as well as pairwise $F_{ST}$ measures suggests that Cartwright (NL) is less similar to the Churchill (MB)-dominated cluster than are the other eastern localities. The eastern interior populations, especially Labrador City (NL) and Kuujjuaq (QC), are a mixture of two genetic clusters, one associated with Churchill (MB) and the other with the eastern localities only.

Churchill (MB) is the most genetically diverse locality sampled among red foxes, regardless of genetic marker or diversity measure, while Kuujjuaq (QC) is intermediate in diversity. The four Labrador localities vary in diversity depending on the marker and the diversity measure. Labrador City (NL) and North West River (NL) tend to be intermediate in genetic diversity, while Cartwright (NL) is generally the least diverse locality followed by Port Hope Simpson (NL). This pattern suggests decreased population sizes or reduced mixing of genetic clusters in these locations. Overall, there is heterogeneity across the landscape, suggesting barriers to gene flow between a northeast coastal cluster and a central interior group, with some level of admixture of the two clusters in the northeastern interior.

This population structure would explain the spread of rabies by red foxes in the past. Several documented outbreaks of rabies have occurred in Labrador over the last 60 years with the majority of cases recorded along the northeastern coast of the province [63], including two outbreaks, one in 1988 and another in 2002–2003, which spread to the northern region of the island of Newfoundland [13]. This disease spread is consistent with the close relationship of foxes along the coastal region as reported in this study and by others [64, 65]. However additional cases also occur periodically far away from the coastline, especially in areas around

Labrador City and northeastern Quebec, and such cases likely reflect disease transmission by the central interior population. Multiple periodic incursions of fox rabies into northern Ontario and western Quebec have also occurred, the first in the 1950s when Arctic rabies first entered the region [11], and later during the early 1990s and 2000s [10, 14, 66], consistent with the spread of disease by the central interior fox sub-population from further north.

While the red fox is clearly capable of spreading the disease into southern regions where sustained virus maintenance has occurred [10, 11], its role in maintaining the disease in northern Quebec and Labrador is less clear. It is apparent that many of the outbreaks in these regions are caused by viral groups also circulating in the Arctic and the evidence from the viral phylogeny suggests that these viruses have been transmitted between Arctic and red foxes in areas of sympatry and then spread in waves southwards in the red fox populations. The identification of both A3-17 and A3-18 viral groups in the Labrador City / Wabush region of western Labrador in 2012 may reflect admixture of the red fox populations evident in that area. Since cases due to the A3-17 viral group were also reported there in 2015, it was thought possible that red foxes within this area could be maintaining the virus throughout this period with disease reporting only during periods of high fox population density when opportunities for human contact were elevated. However, this is not consistent with the phylogeny of the A3-17 variant. The viruses from these two years were located on distinct branches (B and D) of the A3-17 cluster (Fig 3) indicating that the 2015 viruses had not emerged from those recovered in 2012. These transmission patterns contrast with those observed for red foxes in southern Ontario where distinct viral types have circulated exclusively in particular geographical areas over long periods [29, 54, 55, 66]. Overall, the evidence suggests that the sporadic and often cyclical nature of rabies outbreaks along the coastal regions of northern Quebec and Labrador as well as the interior are the result of disease transmission during chance interactions between Arctic foxes and distinct red fox populations dependent upon the fluctuating population densities of these two species. These chance events then dictate the subsequent route or routes of viral spread southwards. Clearly the interactions of rabies susceptible hosts can be an important contributor to rabies persistence and spread as has been shown in southern Ontario where the importance of a secondary host species, the striped skunk (*Mephitis mephitis*), in contributing to viral maintenance has become apparent in recent years [29].

This situation in northern Canada is somewhat distinct from that proposed in Alaska where Arctic fox population structure has been identified and linked to the distribution of the three rabies virus sub-lineages circulating in the state; moreover, as found in this study, rabies is reported in both fox hosts in areas of sympatry, but not in the interior of the state which is populated by the red fox only suggesting that the red fox does not act as a reservoir host in this environment [23, 43].

The distinctiveness of foxes in the Cartwright region, either due to past farming activities or isolation or some combination of the two, is interesting given the nature of the virus identified there. Two isolates recovered in 2002 (group A3-4) were distinct from those moving down along the Labrador coast at the time but much more closely related to viruses spreading throughout the interior and along the north shore of the St Lawrence river in Quebec. While the island of Newfoundland is normally free of fox rabies, an outbreak occurred on the island in 2002–2003. The viral group responsible (A3-13) had previously been recovered from just one Arctic fox specimen in Nunavut in 2002 and was quite distinct from the variants circulating in neighbouring parts of Canada at the time. The lack of any trail linking group A3-13 viruses between these two separate areas of the country is notable and explained by either limited surveillance in the area or possibly the transportation of an infected domestic animal (most likely a dog) from the north to Newfoundland. The latter theory would be supported by the fact that the first case was found in the month of December at a location 250 km south of

where ice bridges usually occur and where no ice bridges to Labrador had existed since May 2002. Furthermore, epidemiological evidence suggested that the disease spread out from where the index case was found rather than suggesting that it had existed undetected on this island since the time of the ice bridges.

This situation does underline the historical challenges of studying wildlife diseases of the far north. Samples of both diseased and healthy trapped foxes have been difficult to come by without provision of financial incentives to trappers, a situation partly overcome in this study through resources provided through the ArcticNet program. Moreover, the high costs of transportation to southern Canada, where routine disease diagnosis is performed, has precluded the submission of many specimens thereby limiting the number of rabies cases identified and hence our understanding of the epidemiology of the disease in the north. These challenges resulted in the limited overlap in specimens available for both the viral and host studies presented here particularly for the earlier years of the study. Notwithstanding these limitations, the combined exploration of host population structure and viral diversity presented in this study has provided new insights into the mechanisms by which rabies is maintained in the far north and subsequently spread into more temperate areas. It should also be acknowledged that over the last decade improved communications via internet access and better education on the risk of rabies in northern communities appears to have increased efforts to monitor the disease in the north and application of the DRIT by many provincial authorities permits identification of additional cases which would otherwise go undetected. Such efforts, together with the insights provided in this report, will improve assessment of the risks to public health posed by rabies in northern communities of eastern Canada and hopefully form the basis of a more informed surveillance network to facilitate early detection of southwards movement of the disease.

## Supporting information

**S1 Fig. Maps showing distribution of rabies virus A3-17 and A3-18 sub-groups across Canada.** The maps were generated as described for Fig 4.
(PNG)

**S2 Fig. Topographical map of eastern Canada showing the proportions of three genetic clusters of red foxes in each locality.** The pie charts of each genetic cluster are as presented in Fig 7A. Dashed lines indicate changes in pattern of genetic variation between northwestern vs northeastern localities (left) and interior and coastal localities (right). The map was generated in topographical-map.com.
(PPTX)

**S1 Table. List of all samples included in rabies virus sequence characterisation.**
(XLSX)

**S2 Table. Characteristics of sites identified by MEME as being under positive selection.**
(DOCX)

**S3 Table. Occurrence and distribution of 18 control region haplotypes among 162 red foxes, including 23 rabies positive animals, across Canada.**
(DOCX)

**S4 Table. Occurrence and distribution of 33 control region haplotypes among 157 Arctic foxes, including 27 rabies positive animals, across Canada.**
(DOCX)

**S5 Table. Characteristics of microsatellite loci used in this study.**
(DOCX)

**S6 Table. Mitochondrial control region haplotypes of the 50 rabies-positives foxes collected from different parts of Canada during the 2012–2013 rabies outbreak.**
(DOCX)

## Acknowledgments

We thank our ArcticNet project collaborators, including Dr. Patrick Leighton, Dr. Audrey Simon, Dr. Emily Jenkins and Dr. Emilie Bouchard, for their co-ordination of sample collection in many northern communities. We also thank the many individuals and agencies involved in sample collection including NL wildlife and conservation officials, trappers Luke Parsons (Labrador City) and Frank Phillips (North West River), and the Nunatsiavut Government. Additional samples, and logistical support, were obtained from passive surveillance provided through the efforts of the Quebec agencies Ministère de l'Agriculture, des Pêcheries et de l'Alimentation du Québec (MAPAQ), Ministère des Forêts, de la Faune et des Parcs (MFFP) and Ministère de la Santé et des Services Sociaux (MSSS), the Centre québécois sur la santé des animaux sauvages, as well as the Newfoundland and Labrador Department of Fisheries and Land Resources (Animal Health Division). We also thank Dr. C. Fehlner-Gardiner and the staff of the National Reference Centre for Rabies at CFIA for provision of samples and meta-data used in this study and R. Duivesteyn for expert assistance in map generation. We are indebted to Dr A. Massé (MFFP) for suggesting improvements to this work and help in ensuring the accuracy of all Quebec sample data. We acknowledge the excellent technical support of G. Vakulenko and A. Hartke for the rabies virus sequencing and the help of S. Predham with mitochondrial DNA sequencing.

## Author Contributions

**Conceptualization:** Susan A. Nadin-Davis, Hugh Whitney.

**Data curation:** Susan A. Nadin-Davis, Emilie Falardeau, Alex Flynn.

**Formal analysis:** Susan A. Nadin-Davis, H. Dawn Marshall.

**Funding acquisition:** Susan A. Nadin-Davis, Hugh Whitney, H. Dawn Marshall.

**Investigation:** Susan A. Nadin-Davis, Emilie Falardeau, Alex Flynn.

**Methodology:** Susan A. Nadin-Davis, Emilie Falardeau, Alex Flynn.

**Project administration:** Hugh Whitney, H. Dawn Marshall.

**Resources:** Susan A. Nadin-Davis, Hugh Whitney, H. Dawn Marshall.

**Supervision:** Susan A. Nadin-Davis, H. Dawn Marshall.

**Visualization:** Susan A. Nadin-Davis, H. Dawn Marshall.

**Writing – original draft:** Susan A. Nadin-Davis, H. Dawn Marshall.

**Writing – review & editing:** Susan A. Nadin-Davis, Emilie Falardeau, Alex Flynn, Hugh Whitney, H. Dawn Marshall.

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
