## [Decision Letter · Decision Letter 0]

23 Oct 2020

PONE-D-20-18736

Relationships between fox populations and rabies virus spread in northern Canada

PLOS ONE

Dear Dr. Nadin-Davis,

Thank you for submitting your manuscript to PLOS ONE. After careful consideration, we feel that it has merit but does not fully meet PLOS ONE’s publication criteria as it currently stands. Therefore, we invite you to submit a revised version of the manuscript that addresses the points raised during the review process.

Both reviewers agree that the manuscript is of interest, but require a number of clarifications. Reviewer 2 in particular has asked for more details on the analyses to better evaluate the results. Please address all reviewer comments prior to re-submission.

We look forward to receiving your revised manuscript.

Kind regards,

Michael Knapp

Academic Editor

PLOS ONE

Journal Requirements:

2. We note that you are reporting an analysis of a microarray, next-generation sequencing, or deep sequencing data set. PLOS requires that authors comply with field-specific standards for preparation, recording, and deposition of data in repositories appropriate to their field. Please upload these data to a stable, public repository (such as ArrayExpress, Gene Expression Omnibus (GEO), DNA Data Bank of Japan (DDBJ), NCBI GenBank, NCBI Sequence Read Archive, or EMBL Nucleotide Sequence Database (ENA)). In your revised cover letter, please provide the relevant accession numbers that may be used to access these data. For a full list of recommended repositories, see http://journals.plos.org/plosone/s/data-availability#loc-omics or http://journals.plos.org/plosone/s/data-availability#loc-sequencing

3. We note that [Figures 1, 4 and S1] in your submission contain [map/satellite] images which may be copyrighted. All PLOS content is published under the Creative Commons Attribution License (CC BY 4.0), which means that the manuscript, images, and Supporting Information files will be freely available online, and any third party is permitted to access, download, copy, distribute, and use these materials in any way, even commercially, with proper attribution. For these reasons, we cannot publish previously copyrighted maps or satellite images created using proprietary data, such as Google software (Google Maps, Street View, and Earth). For more information, see our copyright guidelines: http://journals.plos.org/plosone/s/licenses-and-copyright.

1.     You may seek permission from the original copyright holder of Figures [1, 4 and S1] to publish the content specifically under the CC BY 4.0 license.  

Reviewers' comments:

Reviewer's Responses to Questions

**Comments to the Author**

1. Is the manuscript technically sound, and do the data support the conclusions?

Reviewer #1: Yes

Reviewer #2: Yes

2. Has the statistical analysis been performed appropriately and rigorously? 

Reviewer #1: Yes

Reviewer #2: Yes

3. Have the authors made all data underlying the findings in their manuscript fully available?

Reviewer #1: Yes

Reviewer #2: Yes

4. Is the manuscript presented in an intelligible fashion and written in standard English?

Reviewer #1: Yes

Reviewer #2: Yes

5. Review Comments to the Author

Reviewer #1: This manuscript provides new data and analysis on rabies virus and host genetics in the Canadian North

The paper is very well written and the experiments and analysis are very well described and appropriate to address the question of rabies spread and maintenance in the Canadian North. Conclusions are well supported by the data presented. The findings and discussion contribute to our understanding of rabies dynamics is the far North, and analyses virus spread into more southern regions.

I only have a view minor comments:

Line 89: should this be last century instead of earlier this century? How much has fox farming decreased in the last 20 years vs 50 or 100- years?

Line 701: sentence needs to be corrected

Line 708: Should this be group A3- 4 instead of group 4?

Reviewer #2: This is an interesting study and an impressive amount of work and sampling considering the remote locations being studied. However, I think the presentation could be improved. In particular, more details are needed in the methods, and some of the most interesting findings are presented in text only and are not visible on the figures.

Major comments:

Most of the conclusions about how rabies spreads within and between hosts cannot currently be verified, because host information is missing from key figures. Please annotate the time-scaled phylogeny and the maps with at least the major hosts (e.g. arctic fox, red fox, other).

Line 119: More detail about temporal differences in surveillance is needed, as the data covers a very long time period. I was left wondering for how much of this period non-contact animals had been tested, and how such samples were treated before the DRIT became available. I was under the impression that the DRIT is a relatively recent development?

Line 148-150: More details needed here:

1. Which alignment algorithm was used?

2. Which substitution models were used for these analyses?

3. Three distinct phylogenetic methods are discussed, but the maximum likelihood results are never presented. I don’t think all three are necessarily needed, but arguably the maximum likelihood phylogeny would be more robust than the neighbour joining analysis presented, if both are available.

Line 162: Which phylogeny was used in these analyses? The preceding text describes multiple phylogenetic reconstructions, and here a new alignment is introduced. Since nothing is specified I’d assume the authors relied on automatic reconstructions of phylogeny by HyPhy/datamonkey, but if so this needs to be specified, ideally along with the substitution model used in these instances. Using the same phylogeny as presented elsewhere would be better, but otherwise at least a statement confirming that the topology didn’t differ from what is shown is needed.

Line 237: Which diversity measures were used? A very large number of measures exists (e.g. Shannon, Simpson, etc.), and a quick look at the Arlequin documentation suggests it can calculate several (though the documentation does not specify which).

Line 269: Clade A1 is not strongly supported – the node grouping all A1 sequences has just 51% bootstrap support, and no value is shown for the split between A1 and all other clades.

Line 282: The methods did not contain information about how the outgroup was chosen. Also, more information should be given here:

1. Please annotate major hosts (as discussed above)

2. Which of the tips represent sequences generated in this study?

3. Do clusters correspond to anything (e.g. broad locations)?

Line 319: As above, this figure should show hosts, and here knowing the hosts becomes central to the arguments of the paper.

Line 324: “posterior value” is vague – presumably this should be “posterior probability”?

Line 327: Can the cases with no precise coordinates be illustrated another way, e.g. by writing the number of cases in the centre of Alaska, with the font coloured the same as the group? At present it’s unclear from the maps and other figures where Alaskan samples fit in and whether they are all a single phylogenetic group.

Line 350: Point shapes should reflect sampled host. Potential changes in host as the virus spreads are discussed in the text, so it would be nice to be able to see this and to get a sense of how many cases are being used to support such statements.

Line 373: When giving estimated dates of MRCAs, please include some measure of confidence, such as a 95% credible interval / HPD.

Line 615-618: I could not follow this argument – how can the suggested limited level differentiation be supported by findings of significant differentiation? The following sentence goes on to say that others have found North/South differentiation.

Minor comments:

Line 35-38: It took me a while to understand this sentence and that it is the species which differ, and not the data types giving conflicting results. This might be clearer with an added “both”, as in: “…, but both mitochondrial DNA control region sequences and 9-locus microsatellite genotypes revealed…”

Line 55: Technically this should read “Rabies lyssavirus (RABV)…”, as the sentence is referring to the species.

Line 134: The acronym “AFX” is used throughout the manuscript, but has not been defined as far as I can tell. Since only one variant is being discussed and there are many other acronyms to remember, is this one needed?

Line 194: In figure 1, it is very hard to distinguish the small circles from small squares used to plot “other” locations.

Line 199: Potential typo: should this be “hamstermap.com”?

Line 283: The methods said version 7 of MEGA was used.

Line 372: Unclear why the numbering scheme changes for these lineages, both here and in the plot. Earlier A3 sub-lineages were simply referred to by number.

Line 511: Is this line meant to refer to table S5 instead?

Line 589-590: I found the final part of this sentence confusing, since no results contrasting the rabies and host population genetics results had been presented by this point. Perhaps write: “allowing us correlate…” or “this allows us to correlate”, to indicate that this is still coming in the next paragraphs.

Line 736: Is the word “detection” missing here?

Supplementary document:

1. Caption for table S3 is incomplete, with placeholder text where the authors intended to add accession numbers.

2. Caption for table S4 refers to a “Table d1”, which is presumably a typo.

6. PLOS authors have the option to publish the peer review history of their article (what does this mean?). If published, this will include your full peer review and any attached files.

Reviewer #1: **Yes: **Karsten Hueffer

Reviewer #2: No

---

## [Author Response · Author response to Decision Letter 0]

4 Dec 2020

Please see the response to reviewers file.

---

## [Editor Report · Decision Letter 1]

21 Jan 2021

Relationships between fox populations and rabies virus spread in northern Canada

PONE-D-20-18736R1

Dear Dr. Nadin-Davis,

We’re pleased to inform you that your manuscript has been judged scientifically suitable for publication and will be formally accepted for publication once it meets all outstanding technical requirements.

Kind regards,

Michael Knapp

Academic Editor

PLOS ONE
---

## [Editor Report · Acceptance letter]

3 Feb 2021

PONE-D-20-18736R1 

Relationships between fox populations and rabies virus spread in northern Canada 

Dear Dr. Nadin-Davis:

I'm pleased to inform you that your manuscript has been deemed suitable for publication in PLOS ONE. Congratulations! Your manuscript is now with our production department. 

Kind regards, 

on behalf of

Dr. Michael Knapp 

Academic Editor

PLOS ONE